# Integrative Analysis of Hepatopancreas Transcriptome and Proteome in Female *Eriocheir sinensis* under Thermal Stress

**DOI:** 10.3390/ijms25137249

**Published:** 2024-06-30

**Authors:** Tingshuang Pan, Tong Li, Min Yang, He Jiang, Jun Ling

**Affiliations:** 1Fishery Institute of Anhui Academy of Agricultural Sciences, Hefei 230031, China; pantingshuang@163.com (T.P.); little_li_tong@163.com (T.L.); yangmin831001@163.com (M.Y.); jianghe@aaas.org.cn (H.J.); 2Key Laboratory of Aquaculture & Stock Enhancement in Anhui Province, Hefei 230031, China

**Keywords:** proteome, transcriptome, high temperature, *Eriocheir sinensis*

## Abstract

The Chinese mitten crab (*Eriocheir sinensis*), an economically important crustacean that is endemic to China, has recently experienced high-temperature stress. The high thermal tolerance of *E. sinensis* points to its promise in being highly productive in an aquacultural context. However, the mechanisms underlying its high thermal tolerance remain unknown. In this study, female *E. sinensis* that were heat exposed for 24 h at 38.5 °C and 33 °C were identified as high-temperature-stressed (HS) and normal-temperature-stressed (NS) groups, respectively. The hepatopancreas of *E. sinensis* from the HS and NS groups were used for transcriptome and proteomic analyses. A total of 2350 upregulated and 1081 downregulated differentially expressed genes (DEGs) were identified between the HS and NS groups. In addition, 126 differentially expressed proteins (DEPs) were upregulated and 35 were downregulated in the two groups. An integrated analysis showed that 2641 identified genes were correlated with their corresponding proteins, including 25 genes that were significantly differentially expressed between the two omics levels. Ten Gene Ontology terms were enriched in the DEGs and DEPs. A functional analysis revealed three common pathways that were significantly enriched in both DEGs and DEPs: fluid shear stress and atherosclerosis, leukocyte transendothelial migration, and thyroid hormone synthesis. Further analysis of the common pathways showed that *MGST1*, *Act5C*, *HSP90AB1*, and *mys* were overlapping genes at the transcriptome and proteome levels. These results demonstrate the differences between the HS and NS groups at the two omics levels and will be helpful in clarifying the mechanisms underlying the thermal tolerance of *E. sinensis*.

## 1. Introduction

In the context of biological organisms, temperature is one of the most important environmental stressors, as it affects growth, nonspecific immunity, antioxidant activity, molting, the hepatopancreatic structure, and the energy metabolism of crustaceans [1,2,3]. With global warming in recent years, heat waves and high temperatures have occurred more frequently in China during the summer [4,5]. Crustaceans are ectothermic animals whose body temperature fluctuates with the water temperature [6], and aquatic animals cannot survive when the water temperature exceeds the normal temperature range [1,7,8]. When the water temperature fluctuates, biochemical and gene expression changes occur in crustaceans. The activities of digestive enzymes, except amylase, in mud crabs (*Scylla paramamosain*) gradually decrease with the temperature rising from 27 °C to 37 °C [9]. Catalase activity is significantly lower in the gills of crab (*Carcinusaestuarii*) at 4 °C than at 17 °C and 30 °C, and the hemolymph protein concentration is significantly lower at 30 °C than at 4 °C and 17 °C [10]. High temperatures (35 °C) can suppress antioxidant capacity and promote molting in *S. paramamosain* [3].

*Eriocheir sinensis* has been cultured and consumed in China for a long time, owing to its economic and nutritional value [11], and 815,318 tons are expected to be produced in 2022 [12]. According to Yuan et al. [13], 28 °C~30 °C is suitable for the molting and growth of juvenile *E. sinensis*. However, young *E. sinensis* encounter extremely high or low temperatures during their life stages [14]. The survival, growth, immunity, enzymes, and metabolism of *E. sinensis* may be significantly affected by increasing water temperatures. At 32 °C, both acid phosphatase and alkaline phosphatase in Chinese mitten crabs are significantly upregulated at 12 h and 24 h and significantly downregulated from 72 h to 96 h, and the abundance of the beneficial bacterial significantly decreases [15]. When *E. sinensis* was cultured in a rice field in the summer, the water temperature reached 32 °C or even higher [16]. The water temperature on the surface and at the bottom of the Chinese mitten crab culture pond in the summer can reach 37 °C and 35 °C, respectively [17]. Chinese mitten crab molting can progress normally at 35 °C and is not affected by temperature changes from 28 °C to 35 °C [18]. According to Peng et al. [19], Chinese mitten crabs began to die when exposed to 35 °C for 3 days, and all crabs died when exposed to 40 °C for 10 min. When Chinese mitten crabs were exposed to 35 °C for 24 h, most myocardial fibers lysed, and nuclei and tissue-connective contents in the myocardial layer also reduced [17]. Thus, the water temperature should be controlled in Chinese mitten crab cultures during the summer. Furthermore, high-temperature-tolerant *E. sinensis* breeding is urgently needed for future culturing, which could reduce the mortality caused by high water temperatures.

The hepatopancreas is a multifunctional organ in crustaceans that performs detoxification, metabolism, and immune functions [20]. The total antioxidant capacity of the hepatopancreas of heat-stressed (32 °C) *E. sinensis* showed no significant difference from 0 to 24 h but decreased significantly from 48 to 96 h when compared with that of a control (24 °C) [15]. When crayfish (*Cherax destructor*) was cultured at different temperatures ranging from 10 °C to 30 °C, the aspartate aminotransferase activity, a hepatopancreas damage indicator, had the highest and lowest values at 30 °C and 10 °C, respectively [2]. Previous studies showed that in kuruma shrimp (*Marsupenaeus japonicus*) cold-challenged at 10 °C, most of the amino acids were significantly dysregulated in the hepatopancreas [21]. When *S. paramamosain* was treated at 8 °C, 12 °C, 16 °C, or 20 °C, *hsp60* and *hsp70* were upregulated but *hsp10*, *hsp40* and *hsp90* were downregulated at 8 °C in the hepatopancreas.

Transcriptome and proteome analyses are often employed to analyze differentially expressed genes (DEGs) and differentially expressed proteins (DEPs), respectively, in various biological experiments over long periods of time [17,22,23,24]. According to Pan et al. [17], 2660 upregulated and 1347 downregulated DEGs were detected in heat-stressed (35 °C) *E. sinensis* compared with those in a control group (25 °C). The expression of the *EsTreh* transcript is inhibited when *E. sinensis* is challenged with cold or hot conditions [14]. To date, combined analyses of the hepatopancreas transcriptome and proteome of high-temperature-stressed *E. sinensis* have not been reported. Important DEGs, DEPs, and pathways related to thermal stress were identified by a combined analysis of the transcriptome and proteome.

In the present study, we identified high- and normal-temperature-stressed *E. sinensis*. The transcriptomes and proteomes of HS and NS *E. sinensis* were sequenced and analyzed, and the correlation between the two omics was also analyzed. These results enhance the knowledge of key genes, proteins, and pathways in HS *E. sinensis* and provide genomic and proteomic resources for the molecular breeding of Chinese mitten crabs.

## 2. Results

### 2.1. Transcriptome and Proteome Annotation

Female *E. sinensis* that died after exposure to 38.5 °C or 33 °C for 24 h were identified as HS and NS, respectively. The transcriptome was sequenced using the Illumina sequencing platform, and 200,718,242, and 191,482,932 raw reads were generated for the HS and NS groups, respectively. After low-quality reads were filtered, 199,113,626 and 189,706,812 clean reads were obtained, and the GC content was 50.04% and 52.60% in the HS and NS groups, respectively. The total mapped reads were 174,600,253 and 169,753,006, with ratios of 88.30% and 90.19%, for the HS and NS groups, respectively (Table 1; Appendix A). All read data are available in the NCBI SRA database under the project ID PRJNA1110614.

A data independent acquisition (DIA) analysis of the hepatopancreas of female *E. sinensis* in the HT and HS groups was performed. The DIA combines the advantages of the data-dependent acquisition (DDA) and sequential window acquisition of all optical fragment ions (SWATHs) to quantify the proteome of HT female Chinese mitten crabs. The data were filtered with standards of a 1.0% false discovery rate (FDR) for the precursor and protein thresholds at the peptide and protein levels in thequalitative analysis. After merging the filtered data from the HS and NS groups, 12,214 precursors and 11,655 unique peptides (Appendix A) representing 2717 proteins (Appendix A) were identified. The relative molecular mass of the identified protein was found to be 10–100 kDa, with approximately 17.2% of the relative molecular masses being greater than 100 kDa; 1000 (36.81%) of the protein sequence coverage was less than 5% (Figure 1). Raw proteomic data were deposited in the ProteomeXchange Consortium (http://proteomecentral.proteomexchange.org/) (accessed on 1 March 2024) via the iProX partner repository under the dataset identifier PXD051770.

### 2.2. Identification of DEGs and DEPs

A principal component analysis (PCA) indicated 79.6% and 93.4% variations at the transcriptome and proteome levels, respectively. Inter- and intra-group variations at the transcriptome and proteome levels were demonstrated using PC1 and PC2, respectively (Figure 2). The results of the hierarchical cluster analysis for the DEGs (Appendix A) or DEPs (Appendix A) between the HS and NS groups indicate that the samples in each group were similar and the samples between the two groups could be easily separated.

A total of 3431 DEGs were identified in the HS and NS groups. Among all the DEGs, 2350 were upregulated and 1081 were downregulated in the HS and NS groups (Appendix A). In addition, 161 DEPs were identified in the HS and NS groups using a DIA-based quantitative proteomic analysis. Among all DEPs, 126 were upregulated and 35 were downregulated in the HS and NS groups (Appendix A). 

Correlation analyses were performed between the transcriptomes and proteomes of the two groups at the omics level. A total of 2641 identified genes correlated with their corresponding proteins. However, the Pearson correlation coefficient between the DEPs and DEGs was not high (R = 0.0851) (Figure 3). Among all the correlated genes, 1961 showed no significant differences at the transcriptome or proteome level. A total of 536 genes were significantly different at the transcriptome level (*p* < 0.05), but there were no significant differences at the protein level. There were no significant differences in the 119 genes at the transcriptome level; however, there was a significant difference (*p* < 0.05) at the protein level. Twenty-five genes showed significant differences at the transcriptome and proteome levels, of which five genes were significantly downregulated at both the transcriptome and proteome levels, four genes were upregulated at the transcriptome level and downregulated at the proteome level, 10 genes were downregulated at the transcriptome level and upregulated at the proteome level, and six genes were upregulated at both the transcriptome and proteome levels.

Among all the overlapping genes, 1961 genes showed no significant differences at the transcriptome and proteome levels, 119 genes showed no significant differences at the transcriptome level but showed significant differences at the proteome level, and 536 genes showed significant differences at the transcriptome level but showed no significant differences at the proteome level. Twenty-five genes were significantly differentially expressed at both the transcriptome and proteome levels, of which five were significantly downregulated at both the transcriptome and proteome levels, four were upregulated at the transcriptome level and downregulated at the proteome level, 10 were downregulated at the transcriptome level and upregulated at the proteome level, and six were upregulated at both the transcriptome and proteome levels (Table 2, Appendix A).

### 2.3. Functional Analysis of DEGs and DEPs 

Gene Ontology (GO) terms from DEGs or DEPs with a *p* value of <0.05 were considered significantly enriched by a GO analysis. The top 20 significantly enriched terms for DEGs and DEPs are shown in Figure 4A and Figure 4B, respectively. Most DEGs that were significantly assigned to the biological process (BP) category were rRNA metabolic processes, primary metabolic processes, rRNA processing, metabolic processes, organic substance metabolic processes, organic acid metabolic processes, and small molecule biosynthetic processes. Most DEGs were significantly enriched in GO terms in the cellular component (CC) category, including intracellular anatomical structure, cytoplasm, membrane-bound organelles, organelles, intracellular membrane-boundorganelles, 90S preribosomes, intracellular organelles, nucleoli, and preribosomes. Most DEGs were significantly enriched in molecular functions (MFs), including catalytic activity, small-molecule binding, nucleotide binding, and nucleoside phosphate binding (Figure 4A).

Most of the GO terms enriched for the DEPs assembled in the BP sub-category were chaperone-mediated protein folding, response to heat, cellular response to topologically incorrect proteins, skeletal myofibril assembly, muscle thin filament assembly, protein folding, response to topologically incorrect proteins, response to temperature stimulus, and FtsZ-dependent cytokinesis. Most DEPs assigned to the CC sub-category were polymeric cytoskeletal fibers. Most DEPs assigned to the MF sub-category were associated with structural constituents of the cytoskeleton, unfolded protein binding, chaperone binding, xylulokinase activity, nucleoside triphosphatase activity, ATP hydrolysis activity, pyrophosphatase activity, hydrolase activity, acid anhydrides, phosphorus-containing anhydrides, and ATP-dependent activity (Figure 4B).

Comparative omics analyses between the proteome and transcriptome showed that ten GO terms were enriched in DEGs and DEPs, including chaperone binding, ATP hydrolysis activity, pyrophosphatase activity, hydrolase activity, acting on acid anhydrides, phosphorus-containing anhydrides, hydrolase activity, acting on acid anhydrides, nucleoside-triphosphatase activity, intracellular non-membrane-bound organelle, non-membrane-bound organelle, response to heat, and response to temperature stimulus (Table 3).

The DEGs from the HS and NS groups were subjected to a KEGG pathway analysis, which classified them into 345 pathways, of which 44 were significantly enriched. The 20 most enriched KEGG pathways are shown in Figure 5A (Appendix A). The most enriched pathways were metabolic pathways, tyrosine metabolism, carbohydrate digestion and absorption, thyroid hormone synthesis, selenocompound metabolism, the retinol metabolism, purine metabolism, proximal tubule bicarbonate reclamation, glycolysis/gluconeogenesis, insulin resistance, peroxisome, starch and sucrose metabolism, ribosome biogenesis in eukaryotes, tryptophan metabolism, valine, leucine and isoleucine biosynthesis, pyrimidine metabolism, drug metabolism (other enzymes), endocrine resistance, the insulin signaling pathway, and glycosaminoglycan degradation.

The DEPs between the HS and NS groups were used for the KEGG pathway analysis and classified into 75 pathways, of which 53 were significantly enriched. The most enriched pathways were apoptosis, Salmonella infection, fluid shear stress atherosclerosis, lipid and atherosclerosis, phototransduction–fly arrhythmogenic right ventricular cardiomyopathy, antigen processing and presentation, gastric acid secretion, the Hippo signaling pathway–fly, viral myocarditis, tight junction, adherens junction, phagosome, estrogen-signaling pathway, dilated cardiomyopathy, pathogenic *Escherichia coli* infection, leukocyte transendothelial migration, shigellosis, platelet activation, and hypertrophic cardiomyopathy (Figure 5B, Appendix A).

Comparative analyses of the transcriptome and proteome revealed three common pathways that were significantly enriched in both DEGs and DEPs, including fluid shear stress atherosclerosis, leukocyte transendothelial migration, and thyroid hormone synthesis (Table 4). The KEGG pathway results show that leukocytes and the thyroid play important roles in the high-temperature-stressed Chinese mitten crabs.

The overlapping differentially expressed genes and proteins in the fluid shear stress and atherosclerosis and transendothelial migration pathways are shown in Table 5. Four genes were enriched in these pathways; three genes (*MGST1*, *Act5C*, and *HSP90AB1*) were significantly downregulated at the transcriptome level but significantly upregulated at the proteome level. Only one gene (*mys*) was significantly upregulated at both the transcriptome and proteome levels.

## 3. Discussion

### 3.1. Survival

In this study, female *E. sinensis* began to die when the temperature of the water reached 33 °C; specifically, the first 10crabs that died had undergone heat exposure for 24 h. When the temperature of the water was increased from 33 °C to 38 °C, more than 90% of the crabs died. Finally, when the temperature reached 38.5 °C, with the time elapsed, 10 crabs remained.

Despite the fact that all the crabs used in this study were obtained from the same breeding farm and were of the same age, they exhibited heterogeneity in their temperature tolerances. This outcome could be attributed to the fact that the crabs were at different molting stages and that the prolonged survival of some of the crabs was related to the variation in the immune system dynamics associated with the different stages. This would be in line with the finding that crustaceans exhibit a molting-stage-driven variation in their immune responses to pathogens [25]. The expression levels of the hemocyanin subunit (1, 2, and 5) increased significantly from the post-molt stage to the inter-molt stage and then declined gradually from the inter-molt stage to the ecdysis stage, illustrating that the activity of the hemocyanin-mediated immune defense was highest in the inter-molt stage and lowest in the ecdysis stage in the swimming crab (*Portunus trituberculatus*) [26]. Xu et al. [27] reported that the expression levels of genes related to antimicrobial peptides and antioxidant enzymes in the hepatopancreas of *S. paramamosain* were significantly upregulated at the post-molt and inter-molt stages compared to the pre-molt stage.

### 3.2. Common DEGs and DEPs in Two Omics

Transcriptome and proteome analyses have been used to elucidate mechanisms underlying thermal and cold stress in aquatic animals [28,29,30]. The thermal stress mechanism in Chinese mitten crabs has been described based on the transcriptome in the heart and gills [17,30], proteome in the gills [30], and microbiota in the gut [15]. However, the integration of transcriptome and proteome analyses of the hepatopancreas in high-temperature-stressed Chinese mitten crabs has not yet been reported. 

In this study, RNA-seq and the DIA were used to analyze the mRNA and protein expression levels in HS *E. sinensis*. Combined transcriptome and proteome analyses revealed that the expression levels of some genes and proteins differed between the HS and NS groups. A total of 11,655 unique peptides and 2717 proteins were identified by a proteome analysis in the hepatopancreas of HS *E. sinensis*. A further study showed 3431 DEGs and 161 DEPs between the HS and NS groups. Among all the DEGs and DEPs, 2350 upregulated and 1081 downregulated DEGs and 126 upregulated DEPs and 35 downregulated DEPs were identified between the HS and NS groups. These results indicate that more DEGs and DEPs were upregulated in the HS group. The coefficient value obtained by the correlation analysis of DEGs and DEPs was 0.0851, which is low and similar to that reported previously [24,31]. Twenty-five genes and proteins may be associated with the thermal stress mechanism in *E. sinensis*. Among the 25 genes, 11 showed similar expression trends at both the transcriptome and protein levels. In contrast, 14 genes showed opposite expression trends at the transcriptome and protein levels. When transcripts were upregulated, proteins were downregulated and vice versa. These opposing expression trends and low Pearson correlation values occurred in the two omics between the HS and NS groups, which maybe correlated with post-transcriptional modifications [24]. The protein abundance levels were not consistent with those of the transcripts in the present study, indicating that post-transcripts were present in HS and NS *E. sinensis*. Important DEGs, DEPs, and post-transcriptional mechanisms were determined by an integrative analysis of the proteome and transcriptome in the HS and NS groups, which could be used as candidates for further studies on HS *E. sinensis*. 

### 3.3. Common GO Terms in Two Omics

In this study, 10 GO terms enriched by DEPs and DEGs were obtained by comparative analyses of proteomes and transcriptomes, including “chaperone binding”, “ATP hydrolysis activity”, “pyrophosphatase activity”, ”hydrolase activity, acting on acid anhydrides, in phosphorus-containing anhydrides”, “hydrolase activity, acting on acid anhydrides”, “nucleoside-triphosphatase activity”, “intracellular non-membrane-bounded organelle”, “non-membrane-bounded organelle”, “response to heat”, and “response to temperature stimulus”, which indicated that these GO terms played an important role in HS *E. sinensis*.

Molecular chaperones promote efficient protein folding, minimize toxic aggregation, and maintain proper protein folding under certain stress conditions [32,33,34]. Heat shock proteins (Hsps) are also involved in stress tolerance [35]. There were 23 genes, including three Hsps (*HSC71*, *HSPA1B*, and *HSPA8*) and *HSCB* in the “chaperone binding” GO term. *HSC71* was upregulated when *Dermatophagoides farinae* was stressed at −10 °C, 41 °C, 43 °C, and 45 °C [36]. *HSPA1B* belongs to the HSP70 family, which is upregulated 3.01 to 13.55 times in the summer and winter in cattle and buffalo [37]. *HSPA8* also belongs to the HSP70 family and can act as a biological marker to assess the effects of thermal stress. It is a missense variant that results in low tolerance to thermal stress [38]. A previous study showed that *HSCB* is an important co-chaperone protein that greatly contributes to the stability of the HscB–IscU complex in *Escherichia coli* [39].

The two GO terms were related to heat and temperature challenges in HS *E. sinensis*. The “Response to heat” and “response to temperature stimulus” GO terms contain 46 and 67 genes, respectively, including *eIF2alpha* and *dnaJ* among others. *eIF-2alpha* is a protein synthesis initiation factor, and the phosphorylated *eIF-2alpha* can be upregulated 2~3 times by thermal stress in *Drosophila* [40]. DnaJ is a molecular chaperone of the Hsp40 family that exerts cytoprotective effects by enhancing thermal stress tolerance and signal transduction [41].

The DEGs and DEPs were associated with three common KEGG pathways: fluid shear stress atherosclerosis, leukocyte transendothelial migration, and thyroid hormone synthesis. Further studies showed that all four genes (*MGST1*, *Act5C*, *HSP90AB1*, and *mys*) were significantly upregulated at the protein level in these pathways. Microsomal glutathione S-transferase 1 (*MGST1*) is a member of phase II detoxifying enzymes and can be active at higher temperatures (e.g., 50 °C) [42]. In the present study, the protein level of *MGST1* in HT *E. sinensis* was the highest of the four genes.

## 4. Materials and Methods

### 4.1. Ethics Approval

The use of crabs in this study was approved by the Experimental Animal Welfare and Ethical Committee of the Anhui Academy of Agricultural Sciences (Hefei, China).

### 4.2. Experimental Design

The experiment was conducted at the Fisheries Institute, the Anhui Academy of Agricultural Sciences, in June, 2023. The experimental *E. sinensis* were 14 months old and cultured in an earth pond at a density of 1000 crabs per acre. The water temperature in the cultured earth pond ranged from 10 °C to 31 °C. Healthy female *E. sinensis* (68.7 ± 3.9 g, body weight) were caught by cages from an earth pond at the Fisheries Institute of the Anhui Academy of Agricultural Sciences and acclimated in a 10,000 L plastic tank with water and maintained at 28 ± 0.2 °C. Chinese mitten crabs were fed sinking pellets twice daily. Seven days later, 180 *E. sinensis* were transferred into six 200 L tanks (30 *E. sinensis* per tank) and aerated continuously. During the experiment, the water temperature was elevated from 28 °C at the rate of 1 °C per 24 h. With an elevated water temperature, *E. sinensis* began to die. Then, the water temperature was elevated at a rate of 0.5 °C per 24 h. The first and last ten dying *E. sinensis* from all experimental crabs were selected and designated as the NS and HS groups, respectively. The water in the six experimental tanks was changed quarterly at the same water temperature daily, and no feed was provided during the experimental period. Five hepatopancreas from each group were sampled and separately stored in −80 °C refrigerators for transcriptome and proteome analyses.

### 4.3. Transcriptome Analysis

A TRIzol reagent (Invitrogen, Waltham, MA, USA) was used to isolate the total RNA from ten hepatopancreas. The purity and amount of the total RNA were assessed using theNanoDrop ND-1000 (NanoDrop, Waltham, MA, USA). The total RNA integrity was determinedusingthe Agilent 2100 Bioanalyzer (Agilent Technologies, Palo Alto, CA, USA) and RNase-free agarose gel electrophoresis. Full-length cDNA was constructed using the TruSeq RNA Sample Prep Kit (Illumina, San Diego, CA, USA) according to the manufacturer’s instructions. The resulting cDNA library was sequenced with the Illumina Novaseq6000 (Gene Denovo Biotechnology Co., Ltd., Guangzhou, China), and 150bp paired-end reads were generated. Fastp (version 0.18.0) [43] was used to filter the original data containing adapters or a low-quality base (*q* value ≤ 20) exceeding 50%. After filtering the low-quality reads, clean reads were used to map the *E. sinensis* reference genome (NCBI_ ASM2467909v1). The mapped reads of each sample were assembled using StringTie (http://ccb.jhu.edu/software/stringtie/) (accessed on 1 March 2024) with the default parameters. When the final transcriptome was acquired, StringTie was used toperformtheexpression level by calculating the fragments per kilobase of exons per million mapped fragments (FPKMs). PCA was performed using the value of the relative difference in the transcriptomes to separate the HS and NS groups.

DEGs were selected with a *p* value of <0.05 and a foldchange of >2 using the R package. A GO functional enrichment analysis of DEGs was conducted using DAVID (https://david.ncifcrf.gov/) (accessed on 1 March 2024). The Kyoto Encyclopedia of Genes and Genomes (KEGG) pathway enrichment was analyzed using KOBAS 3.0 (http://kobas.cbi.pku.edu.cn/) (accessed on 1 March 2024). A GO term with a *q* value of <0.05 was defined as significantly enriched. The KEGG pathway with a *p* value of <0.05 was considered significant.

### 4.4. Proteome Analysis

The DIA proteome method was used to analyze the hepatopancreas of female Chinese mitten crabs. The total protein in the hepatopancreas was extracted as described by Wei [44]. The protein content was determined using the bicinchoninic acid (BCA) reagent (Promega, Madison, WI, USA). TheiST Sample Preparation Kit (PreOmics, Martinsried, Germany) was used for protein zymolysis according to the manufacturer’s protocol. The peptide mixture was re-dissolved in buffer A (buffer A: H_2_O, pH 10.0, adjusted with ammonium hydroxide) and fractionated at a high pH using a nanoACQUITY UPLC system (Waters Corporation, MA, USA) connected to a reverse-phase column (OSfLC250, Shanghai Omicsolution Co., LTD, Shanghai, China). The column flow rate was maintained at 2 µL/min, and the temperature was maintained at 30 °C. An EasyPept Frac NANO automatic fraction collection system (OSAP0003, Shanghai Omicsolution Co., LTD) was used to collect the X fraction, and each fraction was dried in a vacuum concentrator for a DDA nano-HPLC–MS/MS analysis. For the DDA nano-HPLC-MS/MS analysis, an UltiMate 3000 (Thermo Fisher Scientific, Waltham, MA, USA) liquid chromatography system was connected to a timsTOF Pro2 (Bruker Daltonics, Billerica, MA, USA). The instrument was operated in the DDA PASEF mode, with 10 PASEF scans per topN acquisition cycle and accumulation and ramp times of 100 ms each. For the DIA nano-HPLC-MS/MS analysis, the equipment and experimental process were like those used forthe DDA nano-HPLC-MS/MS analysis. The DIA data were acquired in the diaPASEF mode. Raw DIA data were processed and analyzed using Spectronaut 18 (Biognosys AG, Zurich, Switzerland) with default settings, and the retention time prediction type was set to dynamic iRT. SpectronAut determines the ideal extraction window dynamically, depending on the iRT calibration and gradient stability. The cutoff levels of the *p* value for the precursors and proteins were 1%. Proteomic PCA was performed to determine the differences in DEPs between the HS and HT groups. DEPs were selected with a foldchange of ≥1.2 and a *p* value of <0.05. GO and KEGG analyses for DEPs were also performed between the HS and HT groups.

### 4.5. Integrated Analysis of the Transcriptome and Proteome

A correlation analysis between the HS and NS groups was performed using R (version 3.5.1) based on changes in the expression of genes and proteins in the transcriptome and proteome. Maps with four quadrants were created to illustrate alterations in gene and protein expression in the transcriptomic and proteomic data, respectively, with the maps showing the quantification and enrichment of genes or proteins in each region. Comparative analyses of the GO function and KEGG pathways in the transcriptome and proteome were performed to identify significant GO terms and KEGG pathways. The commonly expressed genes were also analyzed in the common pathway to identify important genes and pathways in high-temperature-stressed Chinese mitten crabs.

## 5. Conclusions

In this study, a combined analysis of the transcriptome and proteome of the hepatopancreas was performed to provide novel insights into the mechanisms of high-temperature stress in *E. sinensis*. The combined analysis showed that25 genes, including *HSP90AB1*, *MGST1*, *Ppp6c*, and *mys*, were significantly differentially expressed. Ten GO terms were enriched in DEGs and DEPs, namely, chaperone binding, ATP hydrolysis activity, pyrophosphatase activity, hydrolase activity, acting on acid anhydrides, phosphorus-containing anhydrides, hydrolase activity, acting on acid anhydrides, nucleoside-triphosphatase activity, intracellular non-membrane-bound organelle, non-membrane-bound organelle, response to heat, and response to temperature stimulus. Three common pathways were significantly enriched in both the DEGs and DEPs: fluid shear stress and atherosclerosis, leukocyte transendothelial migration, and thyroid hormone synthesis. This study investigated proteomes and transcriptomes and their correlation in female *E. sinensis* under high-temperature stress and will be useful for clarifying the mechanism of thermal stress in *E. sinensis*.

## Figures and Tables

**Figure 1 ijms-25-07249-f001:**
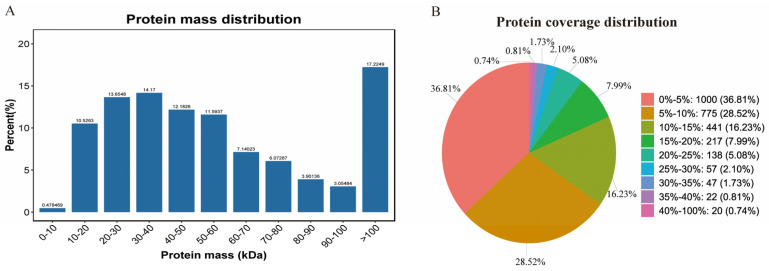
Protein mass distribution (**A**) and protein coverage distribution (**B**). The relative molecular masses of the identified protein were mostly distributed in the range of 10–100 kDa, with about 17.2% of the relative molecular mass being greater than 100 kDa; 1000 (36.81%) of the protein sequence coverage was less than 5%.

**Figure 2 ijms-25-07249-f002:**
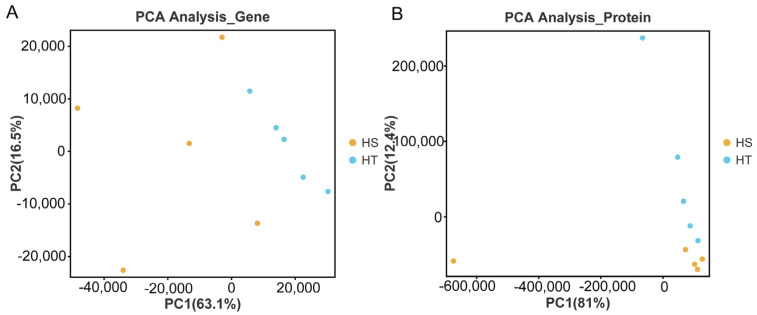
Principle component analysis of transcriptome (**A**) and proteome (**B**) abundance in the HS and NS groups. Red plot indicates that the Chinese mitten crab was obtained from the NS group; blue plot indicates that the Chinese mitten crab was obtained from the HS group.

**Figure 3 ijms-25-07249-f003:**
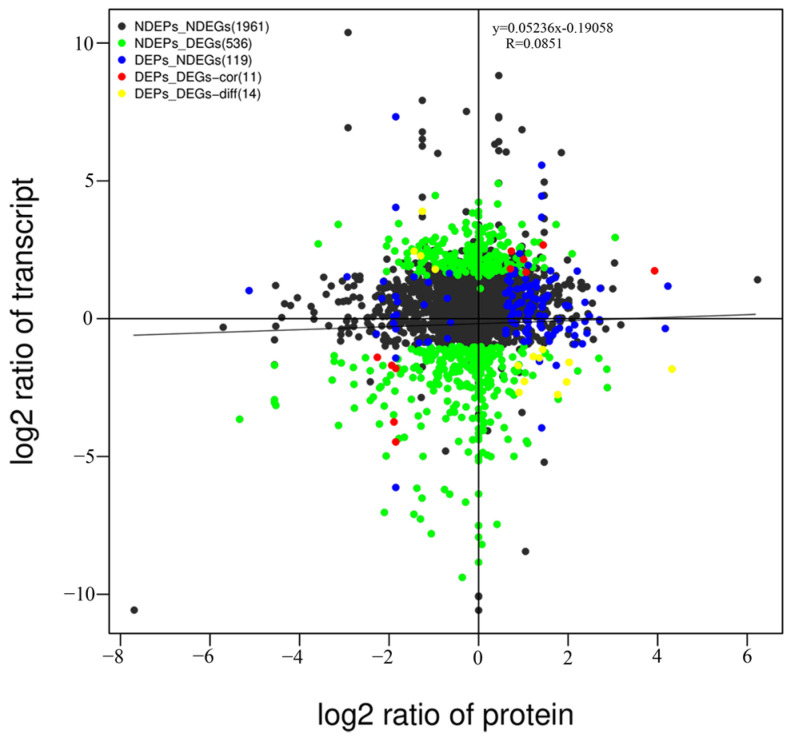
Correlation analyses between the transcriptome and proteome levels. A total of 2641 identified genes correlated with their corresponding proteins. The Pearson correlation between DEPs and DEGs was not high (R = 0.0851).

**Figure 4 ijms-25-07249-f004:**
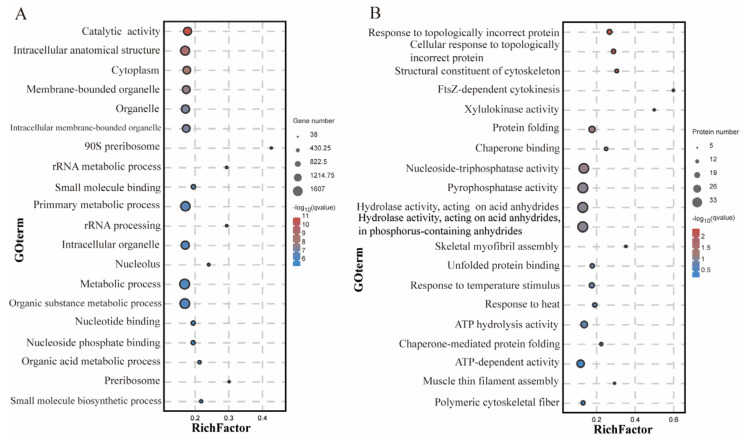
GO enrichment analysis of DEGs (**A**) and DEPs (**B**) between the HS and NS groups. Top 20 significant enriched GO terms. A is the result of the GO enrichment annotation of DEGs, and B is result of the GO enrichment of DEPs.

**Figure 5 ijms-25-07249-f005:**
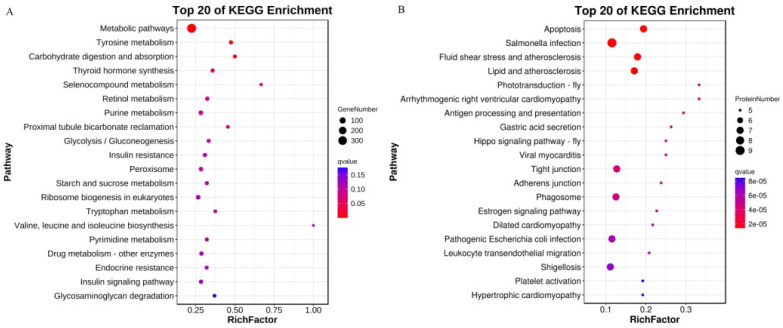
KEGG pathway analyses of DEGs (**A**) and DEPs (**B**) between the high temperature tolerance group and the high temperature sensitive group.

**Table 1 ijms-25-07249-t001:** Summary of Chinese mitten crab transcriptome mapping to the reference genome.

Reads Summary	HS	NS
Raw reads	200,718,242	191,482,932
Clean reads	199,113,626	189,706,812
Raw data	30,107,736,300	28,722,439,800
Clean data	29,761,887,212	28,345,746,874
Valid ratio (reads)/%	99.20	99.07
GC content %	50.04	52.60
Mapping reads (ratio)	174,600,253 (88.30%)	169,753,006 (90.19%)
Unique mapped reads (ratio)	164,765,803 (83.33%)	159,186,593 (84.58%)

**Table 2 ijms-25-07249-t002:** Statistical results of DEGs and DEPs. Among the correlated genes, 25 genes were significantly differentially expressed at both the transcriptome and protein levels, of which 5 genes were significantly downregulated both at the transcriptome and proteome levels, 4 genes were upregulated at the transcriptome and downregulated at the proteome level, 10 genes were downregulated at the transcriptome level and upregulated at the proteome level, and 6 genes were upregulated at the transcriptome and proteome levels.

Expression Pattern	Transcriptome	Proteome	Number of Genes	In Total
Significance of differential expression	Significant	Non-significant	536	2641
	Non-significant	Significant	119	
	Significant	Significant	25	
	Non-significant	Non-significant	1961	
Regulated model	Upregulated	Downregulated	4	25
	Downregulated	Upregulated	10	
	Upregulated	Upregulated	6	
	Downregulated	Downregulated	5	

**Table 3 ijms-25-07249-t003:** Go terms enriched by DEGs and DEPs. Ten terms were enriched by both DEGs and DEPs, including chaperone binding, ATP hydrolysis activity, pyrophosphatase activity, hydrolase activity, acting on acid anhydrides, phosphorus-containing anhydrides, hydrolase activity, acting on acid anhydrides, nucleoside-triphosphatase activity, intracellular non-membrane-bound organelle, non-membrane-bound organelle, response to heat, and response to temperature stimulus.

GO ID	GO Term	GO Function	*p* Value	
			Transcriptome	Proteome
GO:0051087	Chaperone Binding	Molecular function	0.018406	8.07 × 10^−7^
GO:0016887	ATP hydrolysis activity	Molecular function	6.31 × 10^−5^	3.07 × 10^−6^
GO:0016462	Pyrophosphatase activity	Molecular function	0.000202	3.23 × 10^−6^
GO:0016818	hydrolase activity, and acting on acid Anhydrides and phosphorus-containing anhydrides	Molecular function	0.000222	3.36 × 10^−6^
GO:0016817	Hydrolase activity and acting on acid anhydrides	Molecular function	0.000312	3.36 × 10^−6^
GO:0017111	Nucleoside-triphosphatase activity	Molecular function	0.000318	1.61 × 10^−6^
GO:0043232	Intracellular non-membrane-bound organelle	Cellular component	0.006763	8.3954 × 10^−5^
GO:0043228	Non-membrane-bound organelle	Cellular component	0.006787	9.38 × 10^−5^
GO:0009408	Response to heat	Biological process	0.026299	8.92 × 10^−8^
GO:0009266	Response to temperature stimulus	Biological process	0.003881	9.02 × 10^−7^

**Table 4 ijms-25-07249-t004:** Common KEGG pathways were confirmed by DEGs and DEPs. Three pathways significantly enriched by both DEGs and DEPs, including the fluid shear stress and atherosclerosis, leukocyte transendothelial migration, and thyroid hormone synthesis.

Pathway	Pathway Name	*p* Value		Number of Overlapped Genes
		Transcriptome	Proteome	
ko05418	Fluid shear stress and atherosclerosis	0.039719	5.77 × 10^−7^	3
ko04670	Leukocyte transendothelial migration	0.011702	1.48 × 10^−5^	2
ko04918	Thyroid hormone synthesis	0.000732	0.012478	0

**Table 5 ijms-25-07249-t005:** Expression of the overlap genes in the fluid shear stress and atherosclerosis pathways and the leukocyte transendothelial migration pathway at the transcriptome and proteome levels. A total of four genes enriched these pathways; three genes were significantly downregulated at transcriptome level but significantly upregulated at the proteome level. Only one gene was significantly upregulated at both the transcriptome and proteome levels.

Gene	Description	Transcription		Proteome	
		Log_2_FC	Regulation	Log_2_FC	Regulation
*MGST1*	Microsomal glutathione S-transferase 1	−1.83	Down	4.32	Up
*Act5C*	Actin-4	−1.01	Down	1.43	Up
*HSP90AB1*	Heat shock protein 90	−1.38	Down	1.22	Up
*mys*	Integrin	2.458	Up	0.73	Up

## Data Availability

All the data generated or used during this study appear in the submitted article.

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
