# Peer review of "Integrative Analysis of Hepatopancreas Transcriptome and Proteome in Female Eriocheir sinensis under Thermal Stress"

_ijms, 2024, doi:10.3390/ijms25137249_

Round 1

Reviewer 1 Report

Comments and Suggestions for Authors

Comments on the Quality of English Language

the grammar needs to be corrected. For example, line 42: "has been cultured and consumed in china for long time by its economic value and nutrition" - do the authors mean " "has been cultured and consumed in china for a long time because of its economic value and nutrition" ?

Author Response

Reviewer 1

Point 1: This paper is about heat resilience of the Chinese mitten crab. The authors took 180 crabs adapted at 28oC and transferred them in groups of 30 crabs into tanks where the water temperature was elevated at a rate of 1oC per 24 hour or at 0.5oC per 24 hour. Heat tolerant crabs were identified at 38.5oC and heat sensitive crabs at 33oC. This is all the information we are given. This is clearly not sufficient to understand the actual setup of the study.

Response 1: In this study, some Chinese mitten crab cannot tolerant high water temperature at 33 °C and began to die. With the water temperature elevated from 33 °C to 36°C, altogether 10% crabs died. When the water temperature reached 37 °C for 24 h, more than 60% crabs died. When the water temperature reached 38 °C for 24 h, more than 90% crabs died. When the water temperature reached 38.5 °C, with the time elapsed, fewer crabs survived, the last ten survived E. sinensis were designated as the heat tolerant crabs.

Point 2: The GO term enrichment of the proteomics results are not that reliable because only a few were identified (and the raw data is missing - Table 2 only has the identities of the proteins not the number or identity of the peptides identified by LC-MS). These involved folding and purine metabolism which could be interesting. Not surprisingly, the same GO terms were enriched when looking at the overlap since that includes most of the proteins identified. The major problem with this paper is that it describes correlations not causations and even the correlations are not very convincing.

Response 2: A total of 12,214 precursors and 11,655 unique peptides were added and listed in Supplementary Table S2. In this study, we described the correlations of heat sensitive and heat tolerant crabs in transcriptome and proteome levels, and the causations will be further studied in the future.

Point 3: The methods section is not acceptable at all. It is not clear for how long each individual crab was kept at what temperature and what their history was. From the current setup it could be that there are no differences in temperature tolerance but rather there could be another underlying reason such as age or history that allowed some crabs to survive longer. Or it could be coincidence and the DEG/DEP are simply reflecting the time that the crabs were exposed to elevated temperature and are indicative of the temperature stress and not the resilience to it.

Response 3: The experiment was conducted at Fisheries Institute, Anhui Academy of Agricultural Sciences in June, 2023. The experiment E. sinensis were fourteen months old, which cultured in earth pond at the density of 1000 crabs per acre. The water temperature in the cultured earth pond ranged from 10 °C to 31 °C. All the megalpa larva of experiment E. sinensis were from the same breeding farm in Anhui Province. Healthy female E. sinensis (68.7 ± 3.9 g, body weight) were caught by cages from an earth pond in Fisheries Institute of Anhui Academy of Agricultural Sciences, and acclimated in 10000 L plastic tank with following water maintained at 28 ± 0.2 °C.

All the experiment crabs have the same age, and the same heredity background. With the water temperature elevated, some E. sinensis began to die. Water temperature of the first dying E. sinensis from all the experiment crabs was 33 °C, so these crabs were sensitive to high water temperature. Water temperature of the last dying E. sinensis from all the experiment crabs was 38.5 °C, so these crabs were tolerant to high water temperature.

Point 4: The analysis is also not described at all. IT says it was done in R. R is a programming language so this doesn’t give us any information how any of the data was analyzed with the only exception of GO term enrichment.

Response 4: Correlation analysis betweent the HS and HT groups was performed by R (version 3.5.1) based on changes in the expression of genes and proteins in the transcriptome and proteome. Maps with four quadrants were created to illustrate alterations in gene and protein expression in the transcriptomic and proteomic data, respectively, with the map showing the quantification and enrichment of the genes or proteins in each region of the map.

Point 5: Comments on the Quality of English Language. the grammar needs to be corrected.

Response 5: The manuscript has been revised by an experienced English-speaking colleague.

Reviewer 2 Report

Comments and Suggestions for Authors

Dear authors

 While the manuscript is interesting and proof of a sustained effort, it presents a series of weaknesses related to the presentation part.

1. Figures 3 (lines 140-141), 5 (lines 199-201) and 6 (lines 236-237) are written improperly, hard to read and hard to understand by readers. Therefore, I would suggest the authors to keep for the graphic representation only the essential important aspects, with the greatest weight in the context of the work. Secondary or less important elements can be moved to Supplementary Material with a minimal explanation in the text.

Best regards!

Author Response

Reviewer 2

Point 1: Figures 3 (lines 140-141), 5 (lines 199-201) and 6 (lines 236-237) are written improperly, hard to read and hard to understand by readers. Therefore, I would suggest the authors to keep for the graphic representation only the essential important aspects, with the greatest weight in the context of the work. Secondary or less important elements can be moved to Supplementary Material with a minimal explanation in the text.

Response 1: Figures 3 (lines 140-141) has been revised revised as Supplementary Figure S1.

Figures 5 (lines 199-201) has been revised from “Most of the DEPs assigned to the MF sub category were structural constituent of cytoskeleton, unfolded protein binding, chaperone binding, xylulokinase activity, nucleoside−triphosphatase activity, ATP hydrolysis activity, pyrophosphatase activity, hydrolase activity, acting on acid anhydrides, acting on acid anhydrides, in phosphorus−containing anhydrides, and ATP−dependent activity” to “Most of the DEPs assigned to the MF sub category were structural constituent of cytoskeleton, unfolded protein binding, chaperone binding et al.”.

Figures 6 (lines 236-237) has been revised from “The most enriched pathways include Apoptosis, Salmonella infection, Fluid shear stress and atherosclerosis, Lipid and atherosclerosis, Phototransduction - fly, Arrhythmogenic right ventricular cardiomyopathy, Antigen processing and presentation, Gastric acid secretion, Hippo signaling pathway - fly, Viral myocarditis, Tight junction, Adherens junction, Phagosome, Estrogen signaling pathway, Dilated cardiomyopathy, Pathogenic Escherichia coli infection, Leukocyte transendothelial migration, Shigellosis, Platelet activation, and Hypertrophic cardiomyopathy” to “The most enriched pathways include Apoptosis, Salmonella infection, Fluid shear stress and atherosclerosis et al.”

Round 2

Reviewer 1 Report

Comments and Suggestions for Authors

If I understand the author's response correctly, in the end all of the mitten crabs died, which means that even the latest group is not "heat tolerant". Instead of using "heat tolerant" the authors need to describe the state of the animals, e.g. "after heat exposure for x days at x degrees". The authors have also not addressed my concern that there could be additional reasons for prolonged survival, and that the changes observed may be due to the prolonged exposure at elevated temperature not the resistance to it. This needs to be discussed in the manuscript and clearly stated throughout.

Comments on the Quality of English Language

revised English is acceptable

Author Response

Point 1:. If I understand the author's response correctly, in the end all of the mitten crabs died, which means that even the latest group is not "heat tolerant". Instead of using "heat tolerant" the authors need to describe the state of the animals, e.g. "after heat exposure for x days at x degrees".

Response 1: In the article, “heat tolerant” was revised by “thermal stress” or “high-temperature-stress”.

The article name was revised as “Integrative analysis of hepatopancreas transcriptome and proteome in female Eriocheir sinensis under thermal stress”.

In the abstract, “heat tolerant” was also revised as “In this study, female E. sinensis that heat exposure for 24 h at 38.5°C or 33°C were identified as high-temperature-stress (HS) and normal-temperature-stress (NS) groups, respectively.”

Point 2:. The authors have also not addressed my concern that there could be additional reasons for prolonged survival, and that the changes observed may be due to the prolonged exposure at elevated temperature not the resistance to it. This needs to be discussed in the manuscript and clearly stated throughout.

Response 2:

In discussion, additional reason for prolonged survival was stated as follows:

3.1. Prolonged survival

    In this study, female E. sinensis began to die when water temperature reached 33 °C. The first ten dying female E. sinensis were obtained by heat exposure at 33 °C for 24 h. With the water temperature elevated from 33 °C to 38 °C, more than 90% crabs died. When the water temperature reached 38.5 °C, with the time elapsed, fewer crabs survived and the last ten female E. sinensis were obtained.

    All the experiment crabs came from the same breeding farm, and had the same age, but they did not die at the same high-water-temperature. This phenomenon can be speculated that some of the experiment crabs were in different molting stage and the prolonged survival in E. sinensis are related to the immune response at different molting stage. Molting, a phenomenon of shedding the old exoskeleton and re-generating the new one, is one of the most important biological processes of crustacean species. Previously study showed that crustaceans exhibit different immune responses to pathogens at different molting stages [25]. The expression level of hemocyanin subunit (1, 2, and 5) increased significantly from post-molt stage to inter-molt stage and then decline gradually from inter-molt stage to ecdysis stage, illustrating that the activity of hemocyanin-mediated immune defense was highest in the inter-molt stage and lowest in the ecdysis stage in swimming crab (Portunus trituberculatus) [26]. Xu et al. [27] reported that the expression level of genes related to antimicrobial peptide, and antioxidant enzymes in the hepatopancreas of S. paramamosain were significantly upregulated at the post-molt stage and inter-molt stage compared with the pre-molt stage.

Round 3

Reviewer 1 Report

Comments and Suggestions for Authors

I am fine with the current version. 

Comments on the Quality of English Language

The paragraphs added need some English language editing.

Author Response

Comments 1: The paragraphs added need some English language editing. Moderate editing of English language required.

Response 1: The paragraphs added have been revised by an English speaking colleague. All the revised parts in the article were highlighted in red.

The added paragraph was “In this study, female E. sinensis began to die when water temperature reached 33 °C. The first ten dying female E. sinensis were obtained by heat exposure at 33 °C for 24 h. With the water temperature elevated from 33 °C to 38 °C, more than 90% crabs died. When the water temperature reached 38.5 °C, with the time elapsed, fewer crabs survived and the last ten female E. sinensis were obtained.

All the experiment crabs came from the same breeding farm, and had the same age, but they did not die at the same high-water-temperature. This phenomenon can be speculated that some of the experiment crabs were in different molting stage and the prolonged survival in E. sinensis are related to the immune response at different molting stage. Molting, a phenomenon of shedding the old exoskeleton and re-generating the new one, is one of the most important biological processes of crustacean species. Previously study showed that crustaceans exhibit different immune responses to pathogens at different molting stages [25]. The expression level of hemocyanin subunit (1, 2, and 5) increased significantly from post-molt stage to inter-molt stage and then decline gradually from inter-molt stage to ecdysis stage, illustrating that the activity of hemocyanin-mediated immune defense was highest in the inter-molt stage and lowest in the ecdysis stage in swimming crab (Portunustrituberculatus) [26]. Xu et al. [27] reported that the expression level of genes related to antimicrobial peptide, and antioxidant enzymes in the hepatopancreas of S. paramamosain were significantly upregulated at the post-molt stage and inter-molt stage compared with the pre-molt stage.”

The paragraph has been revised as below “In this study, female E. sinensis began to die when the temperature of the water reached 33 °C; specifically, the first 10 crabs that died had undergone heat exposure for 24 h. When the temperature of the water was increased from 33 °C to 38 °C, more than 90% crabs died. Finally, when the temperature reached 38.5 °C, with the time elapsed, 10 crabs remained.

Despite all crabs the used in this study having been obtained from the same breeding farm and being of the same age, they exhibited heterogeneity in their temperature tolerances. This outcome could be attributed to the fact that the crabs were at different molting stages, and that the prolonged survival of some of the crab was related to the variation in immune system dynamics associated with the different stages. This would be in line with the finding that crustaceans exhibit a molting-stage-driven variation in their immune responses to pathogens [25]. The expression levels of the hemocyanin subunit (1, 2, and 5) increased significantly from the post-molt stage to the inter-molt stage and then declined gradually from the inter-molt stage to the ecdysis stage, illustrating that the activity of hemocyanin-mediated immune defense was highest in the intermolt stage and lowest in the ecdysis stage in the swimming crab (Portunustrituberculatus) [26]. Xu et al. [27] reported that the expression levels of genes related to antimicrobial peptides, and antioxidant enzymes in the hepatopancreas of S. paramamosain were significantly upregulated at the post-molt and inter-molt stages compared to the pre-molt stage.”
